# Fungal Disease Tolerance with a Focus on Wheat: A Review

**DOI:** 10.3390/jof10070482

**Published:** 2024-07-13

**Authors:** Akerke Maulenbay, Aralbek Rsaliyev

**Affiliations:** Research Institute for Biological Safety Problems, Gvardeisky 080409, Kazakhstan

**Keywords:** wheat, fungi, tolerance, resistance, host–pathogen interaction, crop protection

## Abstract

In this paper, an extensive review of the literature is provided examining the significance of tolerance to fungal diseases in wheat amidst the escalating global demand for wheat and threats from environmental shifts and pathogen movements. The current comprehensive reliance on agrochemicals for disease management poses risks to food safety and the environment, exacerbated by the emergence of fungicide resistance. While resistance traits in wheat can offer some protection, these traits do not guarantee the complete absence of losses during periods of vigorous or moderate disease development. Furthermore, the introduction of individual resistance genes into wheat monoculture exerts selection pressure on pathogen populations. These disadvantages can be addressed or at least mitigated with the cultivation of tolerant varieties of wheat. Research in this area has shown that certain wheat varieties, susceptible to severe infectious diseases, are still capable of achieving high yields. Through the analysis of the existing literature, this paper explores the manifestations and quantification of tolerance in wheat, discussing its implications for integrated disease management and breeding strategies. Additionally, this paper addresses the ecological and evolutionary aspects of tolerance in the pathogen–plant host system, emphasizing its potential to enhance wheat productivity and sustainability.

## 1. Introduction

Since the latter half of the 20th century, fungal diseases have emerged as significant threats to global agricultural crops. These diseases affect numerous crops identified by the Food and Agriculture Organization (FAO) of the United Nations as essential for human nutrition [1]. Notably, in the 2019 ranking of 137 pests and pathogens, fungi occupied the top six positions for diseases affecting the world’s five most crucial crops [1,2].

Wheat (*Triticum* spp.), identified as one of the world’s most vital crops, ranks third in terms of production and contributes approximately 18% of the global calorie consumption annually. However, the escalating global demand for wheat, projected to increase by 60% by 2050 due to population growth and changing lifestyles, faces challenges due to unforeseen environmental shifts and pathogen movements, posing threats to wheat production [1,3].

In wheat, diseases such as stripe rust (causal agent: *Puccinia striiformis*), leaf rust (*Puccinia triticina*), stem rust (*Puccinia graminis* f. *tritici*), powdery mildew (*Blumeria graminis*), loose smut (*Ustilago tritici*), fusarium head blight (*Fusarium graminearum*), septoria tritici blotch (*Zymoseptoria tritici*), tan spot (*Pyrenophora tritici-repentis*), spot blotch (*Bipolaris sorokiniana*) and the more recent wheat blast (*Magnaporthe oryzae* pathotype *Triticum*) pose significant threats to production, with stripe rust both historically and presently causing substantial losses in susceptible wheat varieties worldwide [2]. Wheat losses caused by septoria tritici blotch and stem rust are of considerable concern, with the current global losses ranging from 5% to 70% [1,4,5].

In response to these challenges, the use of fungicides is often recommended to adjust the dynamics of disease development and the degree of harmfulness [2,6]. However, this approach comes with drawbacks, such as the contamination of the environment and agricultural products [7,8].

The prevalence of fungicides targeting specific cellular sites (monosites) has led to a significant rise in cases of fungicide resistance, often with a surprisingly short time lapse between the fungicide’s commercial release and the subsequent emergence of resistance [6,9,10,11,12]. Approximately 75% of the mode of action groups identified by the Fungicide Resistance Action Committee (FRAC 2022) have documented instances of resistance, presenting a significant challenge to sustainable plant disease management both now and in the future [13].

Methyl benzimidazole carbamates (MBCs), a category of low-use-rate and broad-spectrum fungicides, have been utilized in agriculture for plant disease management since the 1960s [14]. The first documented instance of resistance to MBCs dates back to the late 1960s [15], with over 90 plant pathogens now known to display MBC resistance [9,16]. The introduction of MBC fungicides in cereal crops in the 1970s led to the rapid selection of a resistant allele in the β-tubulin target protein, characterized by a single amino acid substitution (E198A), as indicated by analyses of archived wheat samples [17]. This mutation has become prevalent in *Z. tritici* populations and remains common long after the initial use of MBC fungicides [17]. The Fungicide Resistance Action Committee considers MBC fungicides to pose a high risk for the development of resistance [18].

In the late 1970s, azoles were introduced as popular seed treatments in cereals and for the safeguarding of seed potatoes [19,20]. The onset of azole resistance in plant pathogens was initially observed in 1981 [21]. Azole resistance has been observed in 30 plant pathogens across over 60 countries [9,18]. For *B. graminis*, resistance emerged within four years after the introduction of azoles such as triadimefon and triadimenol in the late 1970s [22]. Other pathogens, including *F. graminearum*, *P. striiformis* and *P. tritici-repentis*, also experienced shifts in sensitivity or developed resistance [18]. The evolution of the resistance to azoles is particularly well documented in *Z. tritici* [19,23,24,25].

Azoxystrobin from Zeneca (now Syngenta) and kresoxim-methyl from BASF were the first strobilurin fungicides (QoIs) introduced to the market in 1996 [20,26]. In 1997, a strobilurin-resistant population of *Plasmopara viticola* was isolated from a field trial site, which led to the definition of a cross-resistance group for these fungicides [27]. In 2002, field isolates of *Z. tritici* resistant to QoIs were identified, all containing a single mutation (G143A) in the mitochondrial cytochrome b protein, which is the target of QoIs. Within two years, this mutation was found to be present in over 80% of field isolates in the UK [28] and had become prevalent across much of Northwestern Europe. The G143A substitution has since appeared in *Z. tritici* populations in other wheat-growing regions worldwide, including the United States [29] and New Zealand [30]. Field resistance also rapidly developed in *B. graminis* and *P. tritici-repentis*, leading to concerns that other cereal diseases could also be at risk [18]. As a result, the use of QoI fungicides as standalone products was discontinued, and they are now used only in mixtures with compounds having a different mode of action [17,31].

Despite the introduction of a new generation of succinate dehydrogenase inhibitors (SDHIs) in 2007, resistant field isolates had been detected in 17 pathogen species by 2017 [9]. Laboratory selection and mutagenesis studies have shown that *Z. tritici* can develop a variety of mutations, some of which confer high levels of resistance to SDHIs [18,32,33]. To maintain the efficacy of SDHIs, they are now used exclusively in mixtures with fungicides that have a different mode of action. During routine monitoring in Europe, field isolates of *Z. tritici* with target site mutations that reduce the sensitivity to SDHIs have been identified. Although it is not yet clear whether these isolates will spread and increase in frequency to a level that compromises disease control, the worst-case scenario suggests that the effective lifespan of SDHIs in managing *Z.tritici* may be limited [17].

This means that there is a constant need for new solutions to control plant-pathogenic fungi [12]. Therefore, the most environmentally sustainable and safe approach involves cultivating agricultural crop varieties equipped with mechanisms to defend against pathogens [9]. The task of comprehending plant defenses against pathogens has been the subject of intensive research in phytopathology for many years. It is now widely recognized that plants employ two primary defense mechanisms against pathogens: the first is resistance, which refers to the host’s capacity to restrict pathogen multiplication, and the second is tolerance, which minimizes the fitness loss of the plant but without reducing the pathogen’s multiplication rate, which denotes the host’s ability to mitigate the negative impacts of infection [34,35]. The outcomes of plant–pathogen interactions can vary significantly depending on whether resistance or tolerance has been initiated. It has been suggested that resistance influences epidemic dynamics by reducing the pathogen’s fitness, thereby exerting selection pressure that could potentially lead to the breakdown of resistance over time [36,37]. In contrast, tolerance does not impose such selection pressure, making it a more stable defense strategy. There is an emerging trend of focusing on tolerance traits in wheat for breeding, demonstrating the value of tolerance over resistance [38]. Stability is a key aspect of the appeal of tolerance as a host trait, as it is less likely to lead to the development of resistance breakdown [39,40,41], which is observed with resistance strategies [36].

Unlike fungicide application, resistant cultivars have relatively low negative environmental impacts, aligning with one of integrated disease management’s (IDM) key objectives: reducing harmful environmental effects. However, these cultivars can exert selection pressure on pathogen populations, potentially leading to resistance. While they might be preferable as a control tactic, they do not inherently offer a sustainable solution for disease management. Even new methods of incorporating resistance into plants may not ensure long-term sustainability.

This challenge highlights the importance of understanding the concepts of “control” and “management” within IDM, which have been extensively studied. “Control” focuses on eliminating pathogens, often resulting in strong selection pressure for resistance. In contrast, “management” involves strategic methods to achieve specific goals. In this context, tolerance, as a form of plant defense, fits well with management, offering appealing features for IDM programs [42,43,44]. 

This review paper aims to provide an overview of the state of the art in relation to tolerance to fungal diseases in agricultural crops, focusing primarily on wheat. Through an analysis of the existing literature, this review explores how tolerance manifests in agricultural settings, quantifies tolerance in wheat through relevant studies and discusses the implications of tolerance to fungal diseases for integrated disease management strategies and breeding practices, emphasizing its significance, challenges and potential to improve wheat productivity. Additionally, this paper delves into the ecological and evolutionary importance of tolerance in the pathogen–plant host system.

## 2. Navigating Challenges in Wheat Disease Resistance: Opportunities Amidst Evolving Pathogens

Plants have evolved intricate defense mechanisms through their long-term co-evolution with pathogens, aimed at safeguarding themselves against diseases. This robust immune system operates via the identification of the key molecular bases of plant immunity—for example, pathogen-associated molecular-pattern-triggered immunity, effector-triggered immunity and quantitative disease resistance [45,46,47,48,49,50]. Despite these sophisticated defense strategies, our understanding of the biotic stress resistance genes in wheat remains limited [3].

Wheat faces significant threats from various pathogens, prompting extensive research into its resistance mechanisms. Notably, over 240 rust resistance genes have been identified, with *Sr31* being a prominent example. However, concerns have arisen with the emergence of the Ug99 stem-rust race, posing a global challenge to wheat production [3,51,52,53]. Similarly, despite the characterization of 84 stripe rust resistance genes [54], many have lost their effectiveness due to the proliferation of virulent races, except for select combinations like *Yr5* and *Yr15*, which remain globally effective [3]. Additionally, 80 leaf rust [55] and 70 powdery mildew [55] resistance genes have been genetically characterized, contributing to race-specific resistance in wheat. Studies on loose smut have revealed simple inheritance patterns, with a limited number of major genes [3]. Moreover, genetic resistance to wheat blast involves five resistance genes, although challenges persist in managing this devastating disease [3,56,57,58,59,60].

Despite the extensive research on genetic resistance in wheat, it remains extremely difficult to achieve durable disease control [61,62]. Introducing individual resistance genes into monoculture exerts selection pressure on the pathogen population [39], leading to the emergence of virulent pathogen strains and compromising the effectiveness of resistance strategies. This phenomenon, known as the “boom–bust” cycle, underscores the limitations of relying solely on resistance mechanisms [63]. In this context, tolerance emerges as a promising alternative, offering a means to mitigate the impact of diseases on yields when complete epidemic control is unattainable [64]. Unlike resistance, which exerts selection pressure on pathogen populations, tolerance is viewed as a potentially sustainable form of disease management [65]. Through our exploration of the concept of tolerance, we can gain insights into novel strategies to enhance crop resilience and sustainability in the face of evolving pathogen threats [36].

## 3. Unveiling the Concept of Tolerance in Plant Pathology: Historical Perspectives and Modern Consensus

The term “tolerance” in plant pathology has a rich historical background, dating back to the pioneering works of Cobb in 1894 and Orton in 1909 [34,66]. These early observations highlighted the ability of certain wheat varieties to maintain high yields despite heavy infection with rust fungi, laying the groundwork for further research into plant tolerance mechanisms [34,66,67]. The seminal study by Caldwell et al., drawing on the earlier research by Salmon and Laude in 1932, Caldwell in 1934 and Newman in 1957, provided an early definition of tolerance as the capacity of susceptible plants to withstand severe pathogen attacks without either significant yield or quality losses [39]. However, despite its significance, tolerance has often been overshadowed in favor of resistance mechanisms.

In the decades following Caldwell’s seminal work, the concept of tolerance remained relatively overlooked in the field of phytopathology [67]. The limited coverage in the research literature and the challenges inherent in the quantitative assessment of tolerance hindered progress in understanding its mechanisms [39,68,69,70]. The confusion surrounding the term’s usage, particularly its distinction from resistance, further complicated the efforts to define and study tolerance effectively.

Recent efforts have sought to refine the definition of tolerance and establish a modern consensus on its conceptual framework [36,62]. It is recommended to begin the process with Schafer’s conceptualization of tolerance as the “capacity of a cultivar resulting in less yield or quality loss relative to disease severity or pathogen development when compared with other cultivars or crops” [67]. This underscores the quantitative and relative nature of tolerance and emphasizes its genetic basis [36]. A diagram illustrating the differences between tolerant and resistant wheat cultivars is shown in Figure 1, which was created using Biorender.com (accessed on 27 June 2024).

In the field of plant pathology, virulence is defined as the degree of negative impact of a pathogen on host fitness components, such as an increase in symptom severity (mortality) and a decrease in biomass production (fecundity) [37,68]. There is an ongoing discussion regarding the significance of plant fitness in cultivated crops, where there is sometimes a misconception equating individual plant fitness with crop enhancement [71]. In the context of virulence, tolerance is emphasized as “the ability of hosts to limit the damage caused by a given parasite burden, which is essentially the ability to minimize per-parasite virulence” [36,62,67,72,73]. In addition, it is crucial to recognize that the assessment of plant fitness can vary depending on the context and the significance of fecundity and mortality in plant–pathogen interactions, leading to a distinction between mortality tolerance and fecundity tolerance [36]. In fact, in the plant–fungus relationship, fecundity tolerance [74,75,76,77,78] has been studied more extensively than mortality tolerance [79]. However, while mortality tolerance has received less attention, it remains an essential aspect in understanding plant–pathogen interactions [75,76,79,80,81,82].

Therefore, when evaluating plants’ tolerance to pathogens among the different major fitness components, such as fecundity or mortality, their various correlates should be considered in relation to the research objectives and the hypotheses under investigation. Failure to do so may result in inconclusive findings regarding the plant’s tolerance to pathogens [36].

## 4. Understanding Tolerance in Host–Pathogen Interactions

Pathogens constitute a significant proportion of Earth’s living organisms, potentially accounting for more than half of all organisms [83,84]. This prevalence suggests that hosts, including plants, continually face challenges throughout their lifespans. This has coincided with the growing recognition of the necessity for a non-anthropocentric perspective on plant disease within non-agricultural ecosystems. In the late 1970s, there arose a need to reformulate the concept of tolerance beyond mere considerations of yield or quality losses [85]. This shift in focus prompted a deeper exploration of tolerance in broader ecological and evolutionary terms, reflecting the intricate interplay between pathogens and their natural hosts within diverse ecosystems [36].

Unlike many trophic interactions among animal species, plants have two primary and distinct defense strategies against natural enemies such as herbivores and pathogens—resistance and tolerance [86]. These two mechanisms can lead to diverse ecological and evolutionary interactions between plants and pathogens [87,88,89,90], influencing the dynamics of plant and pathogen populations differently [62]. Resistance is the most extensively studied defense mechanism of plants against pathogens [62]. Host resistance strategies encompass barriers to infection, mechanisms that swiftly clear infection and processes that restrict the spread of infection within the host. These three types of resistance strategies impede the spread of infection by diminishing the reproductive capacity of the parasite [91]. In contrast, the mechanisms of tolerance traits do not prevent infection itself but mitigate its detrimental effects on the host’s fitness, thereby averting the extinction of plant populations in the presence of high pathogen prevalence [91]. The outcomes of these pathogenic infections can vary widely, ranging from strongly parasitic to commensalistic or even mutualistic interactions between fungi and plants. The results depend on various factors, including the characteristics of both the fungus and the plant involved, as well as the ecological conditions present.

For example, Clay’s new function hypothesis proposes that pathogens can reduce their aggressiveness through the acquisition of new functions that ultimately enhance plant fitness, rather than by simply alleviating the original disease symptoms [92]. Thus, the effects of these interactions can be ambivalent, with the net outcome ranging from intensely parasitic to unconditionally mutualistic.

Early studies examining the metabolic trade-off between tolerance and resistance in host plants have suggested that the evolution of tolerance or resistance to plant damage would be influenced by nutrient availability and the plant’s growth rate [78,90,91,93,94,95,96,97,98]. Studies of the *Syringa vulgaris–Puccinia lagenophorae* system have shown that plant competition affects tolerance and leads to a bimodal distribution of tolerance [75]. Water stress (deficit) reduced the tolerance and competitiveness of plants infected with rust [99], while a low level of available nutrients made plants more tolerant to infection [100]. For example, abiotic stress in *S. vulgaris* genotypes infected with rust often resulted in frost damage, but surviving plants compensated for the losses by increasing the offspring in the next generation [76].

Furthermore, it was suggested that plants with rapid growth rates and shorter lifespans would develop resistance, as the resources lost would be relatively minimal compared to plants with slower growth rates and longer lifespans, for which evolving tolerance would be more beneficial. Shortly after the development of this theory/model, it was proposed that it could be extended to pathogens, including plant pathogens [62,101]. This demonstrates the genetic and phenotypic plasticity of tolerance mechanisms [36]. However, a recent study has provided results that contradict this hypothesis [102].

In general, phytopathogenic microorganisms constitute a common component of any plant community, making them impossible to completely eradicate [36,102].

If the benefits of tolerance outweigh its costs, the allele is fixed by selection in the host plant population; therefore, as the prevalence of the pathogen increases, the advantage of having the tolerance gene also increases [91]. In addition, there is a hypothesis that the tolerance to local pathogens is higher than that to introduced pathogens. For example, the plant *S. vulgaris* is more tolerant to the local fungal pathogen *Coleosporium tussilaginis* than to the introduced pathogenic fungus *P. lagenophorae* [74]. This allows us to consider tolerance in the context of evolutionary biology and the population dynamics of the host plant [36,102]. However, any evidence of pathogen evolution in response to host tolerance is indirect [91]. Restif and Koella used a model to explore how tolerance might influence the evolution of other pathogen traits [90]. They assumed that the host controlled the pathogenic virulence and that virulence and within-host multiplication were positively correlated [103]. The model predicted that an evolutionarily stable state would occur at intermediate levels of host tolerance and pathogen multiplication. Higher or lower levels of pathogen multiplication would disrupt this equilibrium: high levels would lead to host extinction, while low levels would result in the invasion of the host population by a more fecund genotype, potentially leading to pathogen extinction if the invading host genotype was more resistant than the resident tolerant one [90]. Miller et al. demonstrated that pathogens could evolve either higher or lower within-host multiplication rates depending on the nature of the tolerance mechanism [97]. Similarly, van der Bosch et al. [104] found that increased tolerance selected for higher within-host pathogen titers. These models quantified tolerance as either mortality or fecundity tolerance, highlighted the distinction between the two, and noted that mortality tolerance generally had a positive effect on pathogen fitness, with certain exceptions, such as vertically transmitted pathogens or cases where fecundity tolerance came at the cost of a reduced host lifespan [62].

Under agricultural conditions, where the genetic composition of the host population is controlled by humans, studying coevolution is challenging. Under such conditions, only pathogen populations have the opportunity to evolve, while the genetic compositions of host plant populations are subject to artificial selection [36]. Currently, there are insufficient experimental data to confirm the coevolution theory. More experimental research is needed to confirm the impact of tolerance on the evolution of hosts and pathogens.

## 5. Understanding and Assessing Tolerance to Fungal Diseases in Wheat: Methods, Challenges and Future Directions

Various tolerance traits may operate at different organizational levels, spanning from the organ level to the crop level [102,105]. Extensive discussions of the candidate traits linked to tolerance have been covered in prior studies [36,61,62,105,106,107]. Therefore, this section provides only a comprehensive overview. Table 1 presents an extensive summary of the studies assessing the “true tolerance” of wheat cultivars to fungal diseases as the “condition in which two cultivars, exhibiting equal numbers of disease severity at any given time throughout the infection period, show significantly different quantitative responses to the infection” [67]. However, there is evidence of the significant influences of interactions between the genotype and environment (G × E), as well as between the genotype, environment and year of study. This implies that the environment can also influence the extent of tolerance [68,108].

The research shows that some agricultural crop varieties, even those prone to severe infectious diseases, can achieve high yields under favorable conditions. Conditions like high-rainfall seasons or irrigated trials help crops to better realize their yield potential despite disease pressure [38]. For example, phenological and plant-architecture-related traits were not greatly impacted by spot blotch alone. However, these traits significantly declined under terminal heat stress and the combined stress of spot blotch and terminal heat. This indicates that terminal heat and the combination of stresses have a more severe negative effect on these traits compared to spot blotch alone [108].

Additionally, the research underscores that some genotypes show less yield loss than expected across different environmental conditions, suggesting stable tolerance to fusarium crown rot over the years. These genotypes demonstrate high stability and can thrive under various conditions, making them important for breeding programs focused on developing crops with consistent disease tolerance and high yield potential [38].

To effectively study crop tolerance to fungal diseases, a multi-step approach is employed. Initially, a wide range of germplasms is screened to measure changes in yield or growth per unit disease severity. This screening can occur in controlled environments, which offer the advantages of uniform experimental conditions free from both pre-existing levels of the target pathogen and other non-target pathogens for the exploration of tolerance mechanisms and specific trait impacts [110,111,119,121,122].

However, replicating tolerance accurately in controlled settings is challenging due to the complex interplay of various processes influenced by the growth conditions. Therefore, field experiments are crucial in quantifying tolerance [105]. Experiments designed to evaluate the tolerance of genotypes in a cereal breeding program typically involve the replication of both disease-free and diseased plots [124]. Various genotypes are cultivated under these conditions with the objective of identifying genotypes with tolerance, based on the yield reduction observed in the inoculated plots compared to the uninoculated plots. In order to select for genetic tolerance, the assessment method needs to differentiate between the two genetic aspects: the yield potential in the absence of disease and the genotype’s capacity to maintain the yield in the presence of disease [73,126]. The inclusion of a nil disease control method allows the yield potential of cultivars to be estimated [38,72,112,115,116,123].

In reality, in field experiments, completely disease-free plots are difficult to achieve; therefore, it is crucial to use fungicide treatment as the untreated control [39,108,113,114,117,120].

Additionally, achieving consistent disease severity levels across different cultivars or breeding lines is challenging but essential for direct tolerance comparisons. This includes incorporating multiple reference cultivars within each replicate block and considering gradients in variables across the trial area during variance analysis [124]. Several tolerant standard cultivars (2–49, Gala, and Sunco) and intolerant standard cultivars (EGA Bellaroi, Puseas) are typically selected for use in experiments; meanwhile, spatial variation is addressed through appropriate experimental designs [38,124]. Statistical and mathematical models, integrated with experimental data, also help to quantify tolerance and compare varieties effectively [105,118,120,123,127,128,129,130]. This integration helps to identify suitable statistical models to effectively analyze experimental data and address biological questions [105,118].

A methodology was proposed that robustly estimates the relationship between the grain yield and increasing pathogen burdens using response curves [72]. Response curves provide a means of modeling the relationship between the yield and pathogen burden to depict the yield losses due to disease. The basic response curve can be obtained by fitting a linear regression model for the yield against the pathogen burden for each cultivar. The slope parameter estimate from the linear regression model quantifies the yield change per unit increase in pathogen burden, representing the tolerance of a cultivar [78]. Various studies have utilized different models to investigate tolerance traits [105,118,120,123,127,128,129,130]. A schematic overview of the experimental design is shown in Figure 2, which was created using Biorender.com (accessed on 27 June 2024). This figure provides a simplified overview of the experimental design adapted from studies on wheat tolerance to fungal diseases, as highlighted in Table 1. The diagram outlines the key steps and methodologies used to assess the tolerance levels of various wheat cultivars. It includes the stages of pathogen inoculation, disease assessment criteria and data collection methods.

By comprehensively analyzing and comparing the experimental and control plots, tolerant wheat cultivars are identified. These cultivars show significant disease symptoms but do not suffer substantial yield losses, indicating their tolerance to the pathogen.

Although labor-intensive screening methods are common in academic research, they are impractical for the purposes of large-scale screening in plant breeding programs [42,118]. Image-based tools play a significant role in enhancing the disease detection precision [102,131], while bioinformatic tools [130,132] offer valuable insights into pathogen identification and disease prevention [130].

Technological advancements have greatly improved the acquisition and processing of large-scale phenotyping data, previously a significant challenge in functional genomics studies and crop breeding [133]. High-throughput phenotyping platforms now facilitate the measurement of a diverse array of phenotypes, including those related to shoots, roots and canopy and leaf traits [134]. Modern sensor technologies also play a crucial role by providing detailed records of a plant’s environmental history and its responses to changing conditions. Tools such as drones and unmanned aerial vehicles (UAVs), along with devices like pocketPlant3D, equipped with hyperspectral imaging and computed tomography imaging, enable the measurement of traits such as the leaf area index, the detection of weeds and pathogens and yield prediction [133,134]. These technological advancements allow for comprehensive and efficient phenotypic analyses, thereby enhancing crop breeding and functional genomics research. Additionally, computer models and simulation technologies have become essential in modern agriculture and crop breeding. They assist researchers in predicting plant behavior under various conditions, optimizing breeding strategies and understanding the interactions between genetics, the environment and management practices. 

The development of specialized tools, like the OMATEC-HTVP calculator, facilitates the evaluation of host plant resistance vs. tolerance, hybrid vs. modification vigor and pathogen virulence. This calculator is designed to categorize identified variables for easier interpretation and synthesis. It integrates mathematical functions to compare infected plants with uninfected control plants in an experiment. Inspired by the work of Hunt et al. [135], this research strategy aims to effectively select plants with established resistance or tolerance levels against various stresses [136].

Disease severity is assumed in most models of virulence evolution to be positively correlated with within-host pathogen multiplication [36,137,138,139]. For example, the pathogen virulence of *Puccinia triticina* and *Mycosphaerella graminicola* in wheat was positively correlated with pathogen multiplication [140,141]. Recent advances in research have improved the techniques for the quantification of fungal presence in plant tissue, including enzyme-linked immunosorbent assays and polymerase chain reaction (PCR) methods for the measurement of fungal biomass [122,142]. Studies have shown that quantitative PCR (qPCR) results correspond well with visual assessments based on the severity and fungal biomass of stem browning in infected tillers [143,144].

Another strategy to address the requirement for the evaluation of numerous lines while also necessitating detailed measurements could be selecting a limited number of cultivars or mapping population lines that exhibit significant differences in tolerance. These selected lines are then subjected to thorough physiological measurements to pinpoint the key sub-traits associated with tolerance [105]. The level of variability observed in a characteristic is influenced by the size of the population under examination, highlighting the need to assess tolerance in a wide range of cultivars or breeding lines to accurately pinpoint those with high tolerance levels. To facilitate genotypic selection by identifying the quantitative trait loci (QTLs) associated with tolerance or its defining sub-traits, it is crucial to phenotype over 100 lines for subsequent QTL analysis. However, accurately measuring tolerance requires significant resources, which limits the effective screening of cultivars/lines [105,145].

The utilization of molecular markers can help to mitigate these challenges by offering breeders a swift means of ascertaining the presence of specific genes or gene combinations in a breeding line that provide enhanced resistance or tolerance. Recent efforts have been dedicated to identifying markers linked to tolerance traits in a mapping population, streamlining the breeding process for tolerance traits without the need for yield loss testing. Additionally, the Commonwealth Scientific and Industrial Research Organisation (CSIRO) in Brisbane is enhancing existing markers for key resistance QTLs, aiming to enhance their reliability and effectiveness in commercial breeding programs [146].

In the study by Foulkes et al. (2006), the authors aimed to identify cultivars or mapping population lines that exhibited a strong contrast in disease tolerance. Their investigation involved assessing the disease tolerance of near-isogenic lines (NILs) differing in certain traits, such as +awns versus −awns, Rht-D1b (semi-dwarf) versus Rht-D1a (tall) or the presence of the 1BL.1RS chromosome translocation versus its absence (1BL.1BS). The regression analysis revealed that the presence of awns reduced tolerance, while the 1BL.1RS translocation did not notably impact tolerance [113]. Interestingly, the study indicated that the presence of the 1BL/1RS chromosome translocation, known to enhance the radiation use efficiency, appeared to be associated with intolerance [120]. This finding raises the possibility that the lines used in the study, referred to as Hobbit lines, may not have been entirely isogenic. It was hypothesized that QTLs detrimental to the yield could be closely linked to the short awn allele on chromosome 5A, although no such linkage had been previously reported [120].

Resistance mechanisms are typically easier to identify and breed for and are often controlled by a single gene or a major QTL. In contrast, tolerance is primarily influenced by multiple loci and their interactions. Significant QTLs on chromosomes 5A and 6B have been linked to tolerance against snow mold (*Typhula idahoensis*, *T. ishikariensis*, *T. incarnata*) [147] in a biparental population originating from the breeding of the winter wheat varieties “Eltan” [148] and “Finch” [149,150]. Research endeavors should concentrate on improving the genetic resolution of target QTLs to potentially facilitate the discovery and cloning of causal genes, with the goal of gaining a deeper insight into the genetic foundation of tolerance [42].

Recently, a state-of-the art approach, namely a genome-wide association study (GWAS), was applied, exploring the genetic variants statistically associated with disease [151], such as resistance to spot blotch, fusarium head blight and stripe rust [152,153,154,155], as well as important traits in wheat, such as the grain yield [156,157,158], and morphological characteristics like the heading date, plant height and thousand-grain weight [159,160,161]. Additionally, genomic selection has provided promising results in selection for complex traits such as tolerance, and it has been notably validated for snow mold tolerance in Pacific Northwest (PNW) winter wheat [150]. A GWAS pinpointed 100 significant markers spread across 17 chromosomes, with notable single-nucleotide polymorphisms (SNPs) on chromosomes 5A and 5B coinciding with major freezing tolerance and vernalization loci [147]. A larger number of favorable alleles were correlated with improved snow mold tolerance. The selection performed, based on genomic-estimated breeding values and tolerance scores, led to a significant enhancement in tolerance [150]. Compared to traditional biparental QTL mapping, a GWAS offers greater mapping precision owing to the extensive recombination events observed in diverse populations [150].

The combination of genomics, transcriptomics and metabolomics offers a comprehensive understanding of the complex mechanisms governing key agricultural traits, as highlighted by Scossa et al. (2021) [162]. Employing a systems biology approach that integrates various omics datasets, models and predictions of cellular functions is crucial in deciphering the intricate biological processes that underpin complex traits. Integrating multiomics data within a systems biology framework is essential to gain a holistic understanding of dynamic systems, where different biological levels interact with the external environment to express specific phenotypes, as noted by Pazhamala et al. (2021) [163].

## 6. Enhancing Crop Resilience: Exploring Tolerance in Wheat Breeding and Management

### 6.1. Advancements in Genetic Technologies

In the realm of crop breeding and management, there has been a notable shift from the traditional focus on yields towards the recognition of integrated disease management (IDM) strategies, such as the importance of essential genetic traits [43,98,164,165]. The emergence of new breeding methods, such as the revolutionary clustered regularly interspaced short palindromic repeats/CRISPR-associated proteins (CRISPR/Cas) plant genome editing technique, has significantly expanded the plant breeder’s toolkit. This technology has been effectively utilized in nearly 120 crops and model plants, with widespread application in approximately half of them [166]. For instance, in wheat, CRISPR/Cas9 was employed to disable all three TaMLO alleles, resulting in wheat plants with heightened resistance to powdery mildew [167]. Furthermore, recent research has unveiled a novel mechanism through which miRNAs regulate fungal resistance [168]. These advancements underscore the transformative potential of genetic technologies in enhancing crop resilience and sustainability in the face of evolving agricultural challenges.

The future of plant genome editing relies on the widespread adoption and advancement of CRISPR/Cas technology, with a specific emphasis on improving the multiplexing efficiency, refining the high-throughput editing techniques and exploring the potential for chromosomal rearrangements and epigenetic modifications [169].

An illustration of this innovative approach is the German PILTON research project, initiated in 2020, which leverages cutting-edge breeding methodologies to fortify wheat plants with tolerance to fungal pathogens [170]. Central to these advances are key breeding steps that involve targeted mutagenesis using Cas endonucleases, ensuring that the genetic modifications are precisely tailored to specific wheat genes already existing in the wheat genome. Spearheaded by the German Federation of Plant Innovation e.V. (GFPi) and engaging around 60 plant breeding companies, the project is grounded in wheat’s natural pathogen-induced defense reactions. The aim of the research project is to deactivate negative regulators in pathogen defense, which is anticipated to confer broad-spectrum tolerance against various diseases, including wheat leaf rust, stripe rust, septoria leaf blotch and fusarium head blight. Through reductions in the technology’s reliance on specific pathogen types, this approach is poised to establish more universal resilient agricultural practices. The project has transitioned into its fifth stage and is commencing winter wheat trials [170].

However, the existing limited understanding of tolerance raises concerns about how specific environmental factors, such as drought or heat stress, might make tolerant varieties vulnerable to pathogen damage. This issue is particularly relevant in the context of climate change and the increase in drought-prone regions [42,165,171]. Tolerant cultivars must possess agronomically desirable traits to be viable for widespread adoption [42,172]. Understanding the genetic basis and mechanisms underlying plant tolerance presents an opportunity to develop cultivars that can withstand environmental stresses and minimize yield losses [173].

### 6.2. Genetic Strategies for Enhanced Tolerance

To enrich the genetic diversity of the current elite crop pools, unexplored diversity, including crop wild relatives, should be reintegrated into breeding programs to expand the genetic foundation of cultivated crops. This strategy aims to enhance the resilience of cultivated varieties. In the context of anticipated climate change and the shift towards agroecology by 2050, a crucial question arises: which crop’s latent genetic diversity holds the most potential for exploitation [130]?

Crop wild relatives are valuable resources in breeding programs due to their inherent resistance traits against biotic stresses. However, they also possess attributes that enable them to thrive in challenging environments. *Triticum* species, especially those derived from *T. spelta*, have shown promise in enhancing the tolerance to spot blotch, terminal heat and their combination, making them attractive candidates for breeding programs [108].

Over their evolutionary journey, wild progenitors have naturally developed resilience to withstand diverse stresses. In contrast, modern breeding practices, aimed at achieving widespread adaptability, have inadvertently led to a reduction in genetic diversity [174]. There is an opportunity to utilize genetic diversity rather than uniformity in breeding efforts focused on marginal lands [175]. Incorporating the gene pool of wild barley into breeding initiatives not only enriches the genetic diversity via the introduction of novel alleles but also facilitates the reintroduction of lost genes during domestication, thereby enhancing the diversity in breeding programs [174]. Therefore, targeted selection, rather than attempting to include all known resistance genes into a single genotype, has proven to be more effective [174,176,177].

The importance of using plant genetic resources (PGRs) stored in gene banks as sources of beneficial traits and for the preservation of historical diversity is emphasized due to their vital contribution to future improvements in crops. The adoption of novel translocation lines, such as the 1RS.1BL lines sourced from diverse rye varieties, presents encouraging avenues for the enhancement of wheat traits [178]. While the actual application of PGRs in effective crop enhancement has not met all expectations, the use of genomic prediction methods within and among gene banks has successfully identified the best parent combinations for crossing with high-performing cultivars. The resulting offspring from these crosses have exhibited exceptional yield potential in various field trials, outperforming the existing wheat varieties. This well-planned strategy has shown great potential in boosting crop productivity and resilience through its strategic use of genetic resources and advanced breeding technologies [179].

### 6.3. Mixed Cultivar Approaches for Disease Management

Plant breeding strategies can evolve beyond solely seeking resilience or adaptation and instead focus on the capacity for association, such as through intra-specific or inter-specific mixtures, as observed in agroforestry practices [130,180,181,182,183,184]. These mixtures involve cultivating multiple cultivars of the same species simultaneously in a single field, with each cultivar possessing distinct agronomic traits, including disease resistance [61,173]. Therefore, tolerance offers an alternative approach to managing crop diseases and pathogen outbreaks, particularly in situations where complete resistance may not be feasible or practical [61,173]. The effectiveness of cultivar mixtures has been demonstrated across various pathosystems for several decades [185,186,187,188,189,190,191,192,193]. When appropriately selected, mixtures can also enhance product quality [187,191]. Cropping a mixture of susceptible and resistant cultivars in a three-quarter ratio can result in a significant reduction of almost 50% in disease severity, showcasing the effectiveness of incorporating resistant varieties [185].

While susceptible cultivars may offer certain agronomic advantages or be preferred by growers for various reasons, the inclusion of different levels of disease-resistant cultivars in mixtures has demonstrated notable benefits in disease management [185,194,195,196]. However, the recent discovery of neighbor-modulated susceptibility (NMS), which reveals how the susceptibility of a plant can be influenced by the presence of a healthy neighboring plant, has raised some concerns. The prevalence and quantitative impact of NMS in modulating susceptibility in cultivated species remain largely unknown [197]. Growers in the United States have prioritized disease management, not only to increase yields but also to effectively manage the risks [43]. Farmers who opt for tolerant crops may benefit from reduced yield losses, but they may inadvertently contribute to maintaining high infection pressure that can affect neighboring fields [165]. Additionally, private companies may be hesitant to embrace tolerant cultivars to avoid raising doubts about the resistance of their varieties among customers. Therefore, educating stakeholders about tolerance and the benefits of tolerant cultivars is essential for the successful adoption of this management strategy [42]. Traditional mixtures also often consist of cultivars with diverse phenotypes, such as variations in plant size or harvest dates, which can pose challenges in management and incur substantial costs, impeding the widespread adoption of such mixtures [198]. In contrast, multiline mixtures, composed of lines selected for uniformity in their agronomic traits, offer a more streamlined approach to management and adoption [199,200,201,202,203]. Mathematical models have suggested that disease eradication is feasible with an adequate number of varieties in the mixture, with an estimated minimum of approximately five varieties required for eradication. However, due to limitations in genetic resources, complete eradication may not always be achievable, necessitating a larger number of varieties, suggested to be around 10, for effective disease control [186].

## 7. Conclusions

In conclusion, the shift towards prioritizing tolerance over resistance in wheat pre-breeding represents a significant and promising trend in disease management strategies. The stability associated with tolerance as a host trait offers advantages by reducing the likelihood of resistance breakdown, a common issue with resistance strategies. Tolerance provides a valuable alternative approach to mitigate the impact of diseases on yields, especially in situations where complete epidemic control is challenging. Quantifying tolerance as the condition where cultivars with equal disease severity exhibit distinct quantitative responses to infection highlights its potential for integrated disease management and breeding strategies. Embracing tolerance in wheat breeding programs can lead to more sustainable and effective disease management practices, with far-reaching implications for agricultural productivity and resilience.

## Figures and Tables

**Figure 1 jof-10-00482-f001:**
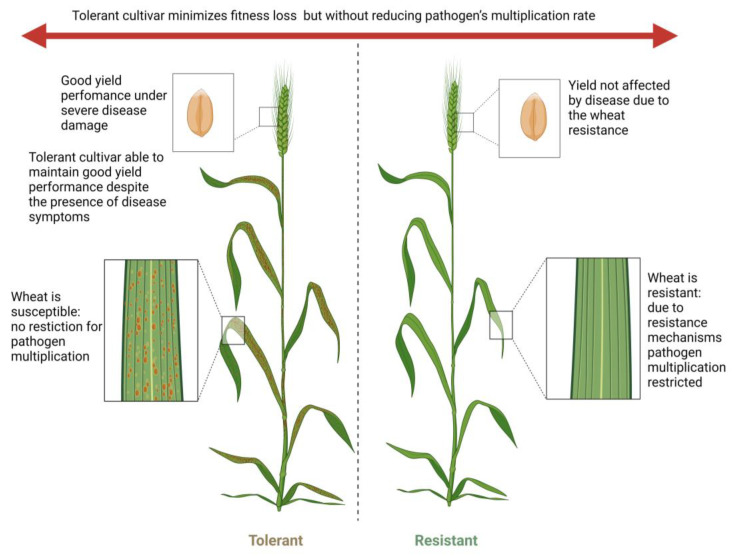
Diagram illustrating the differences between tolerant and resistant wheat cultivars. This figure demonstrates the key differences between tolerant and resistant cultivars in response to pathogen attacks. Resistant cultivars actively combat pathogens through various defense mechanisms, aiming to eliminate or significantly reduce the pathogen’s presence. This often leads to high selection pressure on pathogens, potentially resulting in the development of resistance over time. Conversely, tolerant cultivars do not directly combat pathogens but instead endure their presence while minimizing damage and maintaining productivity. This nonreciprocal response places less selection pressure on pathogen populations, thereby reducing the likelihood of resistance development.

**Figure 2 jof-10-00482-f002:**
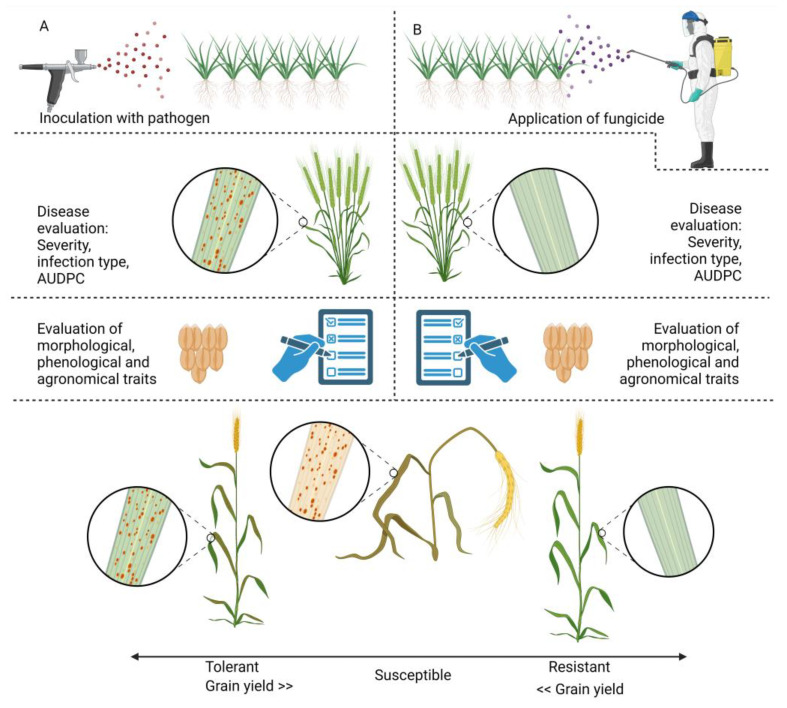
Schematic overview of the experimental design (simplified). (**A**) Experimental plot—inoculation with pathogen: wheat cultivars are exposed to the pathogen; disease evaluation: the disease is assessed using protocols relevant to the study objectives, including severity, infection type, percentage of green leaf area and area under the disease progress curve (AUDPC); key trait evaluation: such as agronomic traits like thousand kernel weight, seed weight and seed quality. (**B**) Control plot (no pathogen exposure)—fungicide application: fungicides are applied to control disease and establish a disease-free plot; disease evaluation: if any, but a healthy wheat cultivar is expected due to fungicide application; key trait evaluation: such as agronomic traits like thousand kernel weight, seed weight and seed quality to evaluate potential yield of healthy wheat cultivar.

**Table 1 jof-10-00482-t001:** Studies of wheat tolerance to fungal diseases highlighting G × E × Y analysis and experimental methodology and relevant references.

Pathogen Species	Genotypes	Field/Greenhouse	Experimental Design	Year of Study	Reference
*Puccinia recondita*	Winter wheat cultivars: **Fulhard (CI 8259)** ^1^, Gladden, Kawvale (CI 8180), Nittany (Pa. 44), Purkoff, Shepherd (CI 6163), Fultz selection (CI 11512)	Field experimental plots (La Fayette, Indiana, USA)	Uninoculated vs. fungicide control	1931	[109]
	Durum and bread winter wheat cultivars: **Fulhard (CI 8527)**, **Kanqueen (CI 12762)**, **Butler (CI 12527)**, Seneca CI 12529, Monon CI 12367, Riley CI 13702, Purdue 579C8, Sel 45	Controlled environmental facilities	Inoculated vs. uninoculated	N/A ^2^	[110]
	Landrace genotypes of *Triticum turgidum* L. ^3^	Field experimental plot (Akaki experimental station, Ethiopia)	Fungicide control vs. no fungicide treatment	1992–1993	[111]
*Puccinia graminis*	Winter wheat genotypes: **GK Bence**, **Bankuti 1201**, **GK Orzse**, Tiszatáj, GK Ságvári, GK Szeged, GK Kincsó, Jubilejnaja 50, GK Órség, GK Bama, GK Kata, SC 79.2567-GK Ságvári, GK Szemes-Quil.xBo-GK Ko, Mapache-GK Kincsó, GK Csörnöc, GK István, GK Zombor	Field experimental plots (Kiszombor station, Hungary)	Inoculated ^4^	1992–1993	[112]
*Puccinia striiformis*	Wheat near-isogenic lines: **Maris Huntsman Rht-D1b/ Rht-D1a**, **Hobbit (spring) unawned**, Hobbit (winter) awned, Weston (1BL.1RS), Chaucer (1BL.1BS)	Field experimental plots (ADAS Terrington, Norfolk, UK)	Non-target disease fungicide control vs. full disease fungicide control	1998–2000	[113]
*Zymoseptoria tritici*	Spring wheat cultivars: Lakish, Bet-Dagan 131, **Miriam**, Mivhor 1177, Yafit	Field experimental plots (Lakhish experimental station, Israel)	Inoculated vs. fungicide control	1971–1974	[114]
	Wheat near-isogenic lines: Mercia Rht-D1b/Rht-D1a, **Hobbit (spring) unawned**, Hobbit (winter) awned, Weston (1BL.1RS), Chaucer (1BL.1BS)	Field experimental plots (ADAS Rosemaund, Herefordshire, UK)	Non-target disease fungicide control vs. full disease fungicide control	1998–2000	[113]
	Wheat genotypes ^3^	Field experimental plots (France)	Fungicide control vs. no fungicide treatment	2006–2011	[115]
	Winter wheat cultivars: Klein Zorro, **Buck 75 Aniversario**, **Buck Brasil**, Buck Guapo, **Klein Escorpion**, **Klein Flecha**, ACA 801, Relmo Centinela, **Nidera Baguette 10**, Klein Chaja	Field experimental plots (Experimental Station Julio Hirschhorn in La Plata, Argentina)	Inoculated ^5^ vs. uninoculated	2010–2011	[116]
	Double-haploid populations of Cadenza and Lynx (UK winter wheat) (C × L): **C** × **L 14B**; Double-haploid populations of LSP2 (Mexican CIMMYT spring wheat of large ear-phenotype) and Rialto (UK winter wheat) (LSP2 × R): **LSP2** × **R 127** and **LSP2** × **R 20**	Field experimental plots (ADAS Rosemaund, Herefordshire, UK and Teagasc, Oak Park, Ireland)	Uninoculated vs. fungicide control	2014–2015	[117]
	Elite wheat cultivars ^3^	N/A	Digital phenotyping approach	N/A	[102]
	N/A	N/A	Mathematical model	N/A	[118]
	Wheat lines: **TRAP#1/BOW**, CROC_1/AE.SQUARROSA (205)//BORL95, **CATBIRD**	Controlled environmental facilities	Inoculated ^5^ vs. uninoculated	2000	[119]
	Winter wheat cultivars: Admiral, Andante, **Avalon**, Beaver, Brigadier, Cadenza, Estica, **Flame**, Galahad, Haven, Hereward, Hornet, **Hunter**, Hussar, Longbow, Lynx, **Mercia**, Norman, Pastiche, Rialto, Riband, Ritmo, Soissons, Spark, Zodiac	Field experimental plots (ADAS Rosemaund, Hereford, UK and ADAS Starcross, Devon, UK)	Uninoculated vs. fungicide control	1995–1997	[120]
	Spring bread wheat: Barkai (YT//NRN/BVR/3/FA/4/CH53//NRN/BVR/3/YQ54/4/2*MERAV), **Miriam (CH53//NRN/ BVR/3/YQ54/4/2*MERAV)**	Controlled environmental facilities	Uninoculated vs. fungicide control	N/A	[121]
*Fusarium pseudograminearum*	Wheat cultivars: **Kennedy**, **Sunco**, Wollaroi	Controlled environmental facilities	Inoculated vs. uninoculated	N/A	[122]
	Wheat genotypes ^6^: **Suntop**, **PBICR-10-002-14**	Field experimental plots (Plant Breeding Institute, Narrabri, NSW, Australia)	Inoculated vs. uninoculated	2015	[123]
	Elite wheat cultivars: EGA Gregory, Lincoln, Sunguard, **Suntop**, **Caparoi**	Field experimental plots (Bullarah, New South Wales, Australia)	Inoculated ^5^ vs. uninoculated	2016	[72]
	Bread wheat cultivars: **ORSS-1757**, **Stephens**, Bauermeister, Bruehl, CT980872, Madsen, CT000161, BURBOT-6, Coda, Weatherford, CT000064, Altay 2000, CT000330, Tubbs 06, Eltan, **2-49**, **Jefferson**, **Gala**, **Tara 2002**, **Sunco**, Macon, Penawawa, CT030799, CT020615, Alpowa, Wawawai, Seri, Calorwa, 302-5, Eden, Otis, Puseas	Field experimental plots (Basin Agricultural Research Center, USA)	Inoculated vs. uninoculated	2000–2008	[124]
	Elite Australian durum breeding material ^6^: **V101030 (JANDAROI /200856)**, **TD1702 (CAPAROI /WID002)**, **V11TD013*3X-63 (WID096/DB ALILLAR OI)**, **V114916 (2-49/EG ABELLARO I (=2/49A30–5)**, **V114942 (2-49/95 0329 (=2/49 B 31–10)**, **Suntop (‘SUNCO’/2* ‘PASTOR ’)/SUN436E)**, **DBA Bindaroi (CAPAROI /261102)**, **Caparoi (LY2.6.3/930054)**, **Jandaroi ((SOURI /WOLLAROI)/KRONOS)**	Field experimental plots (Tamworth Agricultural Institute, NSW, Australia and Liverpool Plains Field Station, UK)	Inoculated vs. uninoculated	2015–2018	[38]
*Bipolaris sorokiniana*	Recombinant inbred lines ^6^ *Triticum spelta* (H + 26) × *Triticum aestivum* (cv. HUW234): **RILs 64**, **71**, **123**, **175**	Field experimental plots (Agricultural Research Farm of Banaras Hindu University, Varanasi, India)	Uninoculated vs. fungicide control	2013–2016	[108]
	Bread wheat genotypes: **Cypress**, Kenyon, Laura and Leader	Field experimental plots (Saskatoon, Saskatchewan, Canada)	N/A	1988–1991	[125]
	Bread wheat genotypes: 1008, ISWYN 32, **Banks**, Hartog, Kite, **Sunstar** and Timgalen	Field experimental plots (Queensland, Australia)	N/A	1989	[125]

^1^ Bold—tolerant genotypes. ^2^ N/A—not applicable. ^3^ Data are unavailable. ^4^ Spreader inoculation—distance from inoculated spreader to disease severity was evaluated. ^5^ Range of pathogen load was used for inoculation. ^6^ Due to the large number of genotypes, a complete list is not provided here; a full list of genotypes can be found in the original paper.

## Data Availability

Not applicable.

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
