# Peer review of "Fungal Disease Tolerance with a Focus on Wheat: A Review"

_jof, 2024, doi:10.3390/jof10070482_

Round 1

Reviewer 1 Report

This review is an excellent paper. I recommend it to be published after with some minor 

My detail comments are inside the attached manuscript 

Author Response

Comment 1: This review is about disease tolerance in wheat, but the author speaks about disease tolerance in several crops. Need to be focused on wheat.

Reply 1: Thank you very much for reading the manuscript and for your valuable comments. We agree with the comment. We have carefully revised our abstract to focus solely on disease tolerance in wheat, as per your suggestion. We have highlighted these changes in the revised manuscript.

Comment 2: There aren´t any examples of fungal disease resistance to fungicides in wheat.

Reply 2: Thank you for your note addressing your concern about the absence of examples of fungal disease resistance to fungicides in wheat. We have now included specific cases of wheat pathogen resistance to fungicides (L64-69; L 72-77; L81-91; L94-101).

Comment 3: This paragraph started with the connector However. It should be changed or the paragraph connected with the previous paragraph (L425 previous version).

Reply 3: Thank you for the comment. We agree and deleted connector however (L619 revised version).

Comment 4: Please don't start the paragraph with a connector. This paragraphs should be joined with the previous (L457) 

Reply 4: Thank you for the comment. We agree and deleted connector however and joined with the previous paragraph (L261).

Reviewer 2 Report

I appreciate some of the author's perspectives, but I also have differing views on certain aspects of this article.

Major concern:

1.        Definitions and comparisons of “tolerance” and “resistance” are not clear, confusing and even conflicting.

2.        The concept of "tolerance" cited in the article is  somewhat abstract, with no reliable metrics to quantify it. The author endeavors to delineate the distinctions between "resistance" and "tolerance." However, the expressions and enumerations provided remain ambiguous. Furthermore, there is an absence of definitive evidence to discern the molecular mechanisms that set "resistance" apart from "tolerance."

3.        Wheat resistance to Fusarium head blight caused by hemibiotrophic fungal pathogen, and no wheat genotype is immune to FHB. Wheat resistance to FHB are quantitative trait and mediated by more than two QTL, and these QTL are believed to confer durable and broad-spectrum resistance by restricting the pathogen multiplication within spike, then should it be classifed into “resistance” or “tolerance”?  

4.        Wheat resistances to biotrophic pathogens, such as rusts and powdery mildew, can be either qualitative or quantitative.

5.        I suggest the author improve the paper through linking “tolerance” to “quantitative resistance (or non race-specific)” and associating “resistance” with “qualitative resistance” (R gene or race-specific resistance), which could be easily understood.

1.              Lines 37-39: keep stripe rust/ yellow rust consistent or use the expression”stripe rust (also called yellow rust)

2.              Line 42 please link “Zymoseptoria tritici and Puccinia graminis” to the diseases, and improve the expressions.

3.              Lines 70-76: I am really confused at the “....while 75 tolerance does not” , what is the mechanism of tolerance?

Author Response

Thank you very much for the comments and suggestions to improve the
manuscript. Please see the attachment. 

Reviewer 3 Report

The authors attempt to review disease tolerance mechanisms in wheat. There is not enough about wheat diseases in this review. It seems like a more general review which the authors point out in their citations that there are several of these general reviews already.

More knowledge of tolerance mechanisms in wheat is needed in this review.

Tolerance is a tricky topic because the definition is not always clear or part of a consensus, although authors make good attempts at explaining conflicting definitions.

The abstract should better reflect the content of the subsections of the review.

Ln 169: Something is missing in these sentences.

Ln 250: If the topic is about wheat, do not use non-wheat examples. This seems to be avocado, but the plant species is not even stated.

Ln 287: again the example is not about a wheat disease. Use wheat disease examples.

Ln 298: GWAS is not cutting edge anymore?

Ln 349: “innate defense mechanisms triggered by pathogen encounters” if it is this, it that not a type of resistance mechanism and not tolerance?  if innate immunity leads to a reduction in pathogen number e.g. plant senses chitin and makes chitinases to attack pathogen as part of innate immunity.

How is technology being used to reduce the costs of studying tolerance in the field? And replicating field conditions in the glasshouse?

A diagram to try to illustrate he difference between tolerance and resistance would be useful. Of course, using a wheat plant in the diagram.

Author Response

Thank you very much for the comments and suggestions to improve the
manuscript. We especially grateful for English corrections in the text.

Reviewer 4 Report

Manuscript entitled “Fungal Disease Tolerance in Wheat: A Review”. The manuscript reviewed the way to understand, access and enhance the tolerance of wheat to fungal diseases. This review provides new insights to deal with fungal disease of wheat. Several points need to be addressed before it can be accepted.

The aim of this manuscript was focus on wheat, but not wildly crop plants. Therefore, examples in this review should relate to wheat.

Some section titles need revised.

“4. Understanding and Assessing Tolerance to Fungal Diseases in Crop Plants: Methods, Challenges, and Future Directions” Crop plants should replace by wheat.

“5. Enhancing Crop Resilience: Exploring Tolerance in Crop Breedingand Management” Crop should replace by wheat.

Section “6. Understanding Tolerance in Host–Pathogen Interactions” should move before section 4.

Manuscript entitled “Fungal Disease Tolerance in Wheat: A Review”. The manuscript reviewed the way to understand, access and enhance the tolerance of wheat to fungal diseases. This review provides new insights to deal with fungal disease of wheat. Several points need to be addressed before it can be accepted.

The aim of this manuscript was focus on wheat, but not wildly crop plants. Therefore, examples in this review should relate to wheat.

Some section titles need revised.

“4. Understanding and Assessing Tolerance to Fungal Diseases in Crop Plants: Methods, Challenges, and Future Directions” Crop plants should replace by wheat.

“5. Enhancing Crop Resilience: Exploring Tolerance in Crop Breedingand Management” Crop should replace by wheat.

Section “6. Understanding Tolerance in Host–Pathogen Interactions” should move before section 4.

Author Response

Thank you very much for reading the manuscript and for your valuable comments. 

We have made all changes according to your comments. 

Round 2

Reviewer 2 Report

This revised version has been significantly improved when compared with the previous version. The authors have addressed my major concerns. 

I recommend publication. 

Author Response

Thank you once again for your valuable contribution to our manuscript.

We greatly appreciate your thorough analysis and the constructive feedback you provided. 

Your invaluable comments and suggestions have undoubtedly contributed to enhancing the quality and clarity of our work.

Reviewer 3 Report

The authors should agree to make the reviewers comments and the authors replies published alongside their manuscript.

Fig. 2. AUDCP => AUDPC?

Line 171: the heading format has merged with the preceding paragraph.

Line 468: “approach” and “was applied”

If there is not so much on wheat, the title could be changed to something like “Fungal Disease Tolerance with a focus on Wheat: A Review”

Add the references in the reply about how tech is being used to reduce costs?

Yang, W., Feng, H., Zhang, X., Zhang, J., Doonan, J. H., Batchelor, W. D., Xiong, L., & Yan, J. (2020). Crop phenomics and high-throughput phenotyping: Past decades, current challenges, and future perspectives. Molecular Plant, 13, 187–214. https://doi.org/10.1016/j.molp.2020.01.008

Jin, X., Zarco-Tejada, P., Schmidhalter, U., Reynolds, M. P., Hawkesford, M. J., Varshney, R. K., Yang, T., Nie, C., Li, Z., Ming, B., Xiao, Y., Xie, Y. & Li, S. (2020). High-throughput estimation of crop traits: A review of ground and aerial phenotyping platforms. IEEE Geoscience and Remote Sensing Magazine, 9, 200–231. https://doi.org/10.1109/MGRS.2020.2998816

see above

Author Response

Thank you once again for your valuable contribution to our manuscript.
